# Improving Neural Language Generation with Spectrum Control

**Lingxiao Wang**[1], **Jing Huang**[2], **Kevin Huang**[2], **Ziniu Hu**[1], **Guangtao Wang**[2], **Quanquan Gu**[1]

[1]Department of Computer Science, University of California, Los Angeles
[2]JD AI Research, Mountain View, CA 94034
`{lingxw,bull,qgu}@cs.ucla.edu`
`{jing.huang,kevin.huang3,guangtao.wang}@jd.com`

## Abstract

Recent Transformer-based models such as Transformer-XL and BERT have achieved huge success on various natural language processing tasks. However, contextualized embeddings at the output layer of these powerful models tend to degenerate and occupy an anisotropic cone in the vector space, which is called the representation degeneration problem. In this paper, we propose a novel spectrum control approach to address this degeneration problem. The core idea of our method is to directly guide the spectra training of the output embedding matrix with a slow-decaying singular value prior distribution through a reparameterization framework. We show that our proposed method encourages isotropy of the learned word representations while maintains the modeling power of these contextual neural models. We further provide a theoretical analysis and insight on the benefit of modeling singular value distribution. We demonstrate that our spectrum control method outperforms the state-of-the-art Transformer-XL modeling for language model, and various Transformer-based models for machine translation, on common benchmark datasets for these tasks.

## 1 Introduction

Neural language generation (NLG) is an important task with many practical applications, such as automatic speech recognition (Graves et al., 2013; Toshniwal et al., 2018), text generation (Bowman et al., 2016; Radford et al., 2019; Keskar et al., 2019), machine translation (Bahdanau et al., 2015; Vaswani et al., 2017) and dialog systems (Gao et al., 2019a; Tang et al., 2019). Most NLG models utilize a complex encoding model to map a given context into a hidden state vector, and then predict the next word distribution by multiplying the encoded vector with the output embedding layer, followed by a softmax layer. In the past few years, it has witnessed a significant progress in NLG by improving the encoding model, from the recurrent neural network (RNN) (Bahdanau et al., 2015; Jozefowicz et al., 2016; Merity et al., 2018a) based models to the current Transformer-based models (Vaswani et al., 2017; Devlin et al., 2019; Dai et al., 2019; Radford et al., 2019). However, embeddings in the softmax output layer have been shown not capable enough to model the conditional probability (Yang et al., 2018).

Recently, Gao et al. (2019b) pointed out another limitation of the output embeddings: the *representation degeneration* problem. They showed that the singular value distribution of the output embedding matrix tends to decay very fast, and the embedding space is squeezed into a narrow cone (as shown in Figure 1(a) and 1(c) in 2-D plots). Such anisotropic shape (Ethayarajh, 2019) is very different from what one would expect from an expressive word embedding space (Arora et al., 2016a; Mu & Viswanath, 2018). Therefore, several efforts (Gao et al., 2019b; Wang et al., 2019a) have been made to address the degeneration problem.

Unlike previous approaches that applied implicit regularization to singular values of the output embedding matrix, we propose a spectrum control (SC) approach, which was inspired by the spectral control technique used for Generative Adversarial Network (GAN) training (Jiang et al., 2019), to explicitly control the singular value distribution. We first reparameterize the output embedding matrix $\mathbf{W}$ by its singular value decomposition (SVD): $\mathbf{W} = \mathbf{U}\boldsymbol{\Sigma}\mathbf{V}^{\top}$, where $\mathbf{U}, \mathbf{V}$ are column orthonormal matrices, and $\boldsymbol{\Sigma}$ is a diagonal matrix of singular values. Then we guide the training of $\boldsymbol{\Sigma}$ by a predefined slow-decaying prior distribution, such as a polynomial decay distribution, or

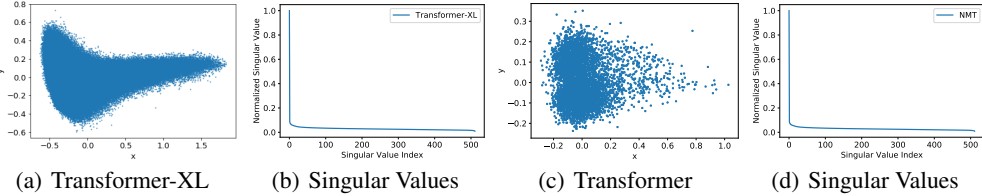

|  |  |  |  |
| (a) Transformer-XL | (b) Singular Values | (c) Transformer | (d) Singular Values |

Figure 1: Projected word embeddings[1]and singular value distributions. (a) and (c): 2-D visualization of word embedding matrices of Transformer-XL for language modeling and Transformer for machine translation; (b) and (d): Normalized singular value distributions of embedding matrices.

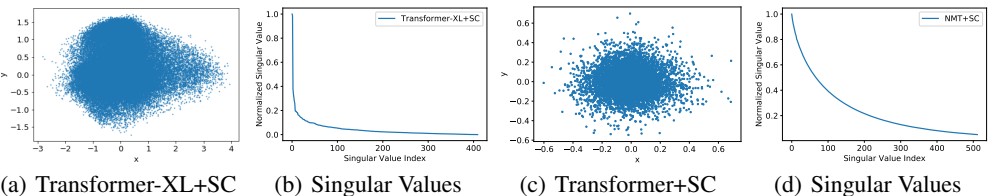

|  |  |  |  |
| (a) Transformer-XL+SC | (b) Singular Values | (c) Transformer+SC | (d) Singular Values |

Figure 2: Projected word embeddings and singular value distributions using our spectrum control method. (a) and (c): 2-D visualization of word embedding matrices of Transformer-XL for language modeling and Transformer for machine translation; (b) and (d): Normalized singular value distributions of embedding matrices.

an exponential decay distribution. At the end of training, the distribution of singular values of the embedding matrix gets close to the prior distribution. Our spectrum control approach alleviates the *representation degeneration* problem by encouraging the diversity of word representations and improving isotropic property of these representations (see Figure 2), even on top of the powerful Transformer-based models.

We further present a theoretical analysis to justify our method. The results suggest that it is beneficial to directly guide the singular value distribution of the embedding matrix throughout the training process and control the decay rate of these singular values. We demonstrate the effectiveness of our training framework with extensive experimental results on two tasks: language modeling and machine translation. Our spectrum control method outperforms the latest state-of-the-art Transformer-XL model on *WikiText-103* dataset for language modeling; and obtains close to $1.5$ BLEU improvement on *IWSLT 2014 German-English* translation task compared to the Transformer baseline model.

## 2 RELATED WORK

In this paper, we mainly focus on the output embedding matrix that is used in the softmax output layer for language generation tasks. This embedding matrix is also used as input word embeddings, which is known as weight tying trick. The weight tying not only reduces the number of parameters but also enjoys theoretical benefits (Inan et al., 2017). Thus the weight tying has been successfully applied to many state-of-the-art models for language modeling and machine translation (Merity et al., 2018a; Vaswani et al., 2017; Yang et al., 2018).

However, there are certain issues with the softmax output layer. Yang et al. (2018) first identified a problem called "softmax bottleneck" that the softmax output layer does not have enough capacity to model natural language due to its connection with the rank bottleneck in matrix factorization. A simple and effective method called Mixture of Softmaxes (MoS) was proposed to deal with this issue. Follow-up work in Kanai et al. (2018); Ganea et al. (2019) tried to replace softmax with alternative activation functions. Kanai et al. (2018) proposed to use *sigsoftmax*, which is composed of a multiplication of an exponential function and sigmoid function. Ganea et al. (2019) proposed a Linear-Monotonic-Softmax (LMS) model that generalized the approach in Kanai et al. (2018) by learning parametric point-wise increasing functions to optimally distort the logits before feeding them into the softmax layer. Pappas & Henderson (2019) instead resorted to a powerful deep

---

[1]Note that we project the original word embeddings to a 2 dimensional vector space using principal component analysis (PCA) for the purpose of visualization.

residual nonlinear output mapping while using a single softmax function without modifying its dimensionality or rank.

Another line of work focuses on increasing the expressive power of the output embedding matrix by adding a cosine similarity regularization term (Gao et al., 2019b), or adversarial noise (Wang et al., 2019a). Gao et al. (2019b) analyzed the *representation degeneration* problem that the output embeddings tend to degenerate and be distributed into a narrow cone. A regularization term based on the summation of pairwise cosine similarity among all words was proposed to increase the representation power of word embeddings. Wang et al. (2019a) pointed out the computation of regularization term in (Gao et al., 2019b) depends on the size of the vocabulary and hence is costly. Instead, they proposed a simple yet highly effective adversarial training method that adds adversarial noises to the output embedding layer when training the models. They proved in theory that this adversarial training increases the distances between two different words, and thus encourages the diversity of the embedding vectors. Our work follows this line of thought, but takes a different approach: motivated by the spectrum control method for GAN training (Jiang et al., 2019), we propose to directly guide the singular values by a slow-decaying prior distribution during the model training process. We show that the anisotropic behavior of the contextualized word representations from powerful Transformer-based models (Ethayarajh, 2019) are alleviated by our spectrum control approach.

## 3 PROBLEM SETUP

In this section, we briefly introduce the neural models for language generation, and illustrate the singular value decay phenomena of existing neural language models. We first introduce some notations used in the rest of the paper.

**Notation:** For a $d$-dimensional vector $\mathbf{x} \in \mathbb{R}^d$, we use $\|\mathbf{x}\|_q = (\sum_{i=1}^d |x_i|^q)^{1/q}$, where $0 < q < \infty$ to denote its $\ell_q$-norm, and $\|\mathbf{x}\|_\infty = \max_i |x_i|$ to be its infinity norm. For a matrix $\mathbf{A} \in \mathbb{R}^{d_1 \times d_2}$, let $\mathbf{A}_{i*}$ be the $i$-th row of $\mathbf{A}$, and we use $\|\mathbf{A}\|_2, \|\mathbf{A}\|_F, \|\mathbf{A}\|_1$ to denote its spectral norm, Frobenius norm, and matrix 1-norm. Given two sequences $\{a_n\}$ and $\{b_n\}$, if there exists a constant $0 < C < \infty$ such that $a_n \leq Cb_n$, we write $a_n = O(b_n)$, and we use $\widetilde{O}(\cdot)$ to hide the logarithmic factors.

### 3.1 NEURAL LANGUAGE GENERATION

We first briefly review the softmax output layer typically used in neural language generation models. We define the joint probability of a given length-$n$ sentence of words (tokens) $\mathbf{s}_n = (\mathbf{y}_1, \ldots, \mathbf{y}_n)$ as the following product of conditional probabilities

$$\mathbb{P}(\mathbf{s}_n) = \prod_{t=1}^n \mathbb{P}(\mathbf{y}_t|\mathbf{c}_t), \tag{3.1}$$

where $\mathbf{y}_t \in \mathcal{V}$ is the $t$-th word in sentence $\mathbf{s}_n$, and $\mathcal{V}$ represents the word vocabulary, $\mathbf{c}_t = \mathbf{y}_{1:t-1} = (\mathbf{y}_1, \ldots, \mathbf{y}_{t-1})$ is referred to as the context of word $\mathbf{y}_t$. In addition, the context $\mathbf{c}_t$ is usually modeled by a fixed size vector $\mathbf{h}_t \in \mathbb{R}^d$, which is referred to as the hidden state, using some neural networks such as LSTM (Hochreiter & Schmidhuber, 1997) and Transformer (Vaswani et al., 2017). Then, the probability distribution of the output word $\mathbf{y}_t$ given the context $\mathbf{c}_t$, i.e., $\mathbb{P}(\mathbf{y}_t|\mathbf{c}_t)$ in (3.1), is parameterized as the following softmax function:

$$\mathbb{P}(\mathbf{Y}_t = \mathbf{y}_t|\mathbf{c}_t) = \mathbb{P}(\mathbf{Y}_t = \mathbf{y}_t|\mathbf{h}_t) = \frac{\exp(\mathbf{h}_t^\top \mathbf{W}_{\mathcal{I}(\mathbf{y}_t)*})}{\sum_{i=1}^N \exp(\mathbf{h}_t^\top \mathbf{W}_{i*})}, \tag{3.2}$$

where $\mathbf{W} \in \mathbb{R}^{N \times d}$ is the weight matrix and is usually tied with the input word embedding matrix (Press & Wolf, 2017; Inan et al., 2017), $N = |\mathcal{V}|$ is the vocabulary size, $d$ is the embedding dimension, $\mathcal{I}(\mathbf{y}_t)$ represents the index of word $\mathbf{y}_t$ in vocabulary $\mathcal{V}$. In the following discussion, we call output weight matrix $\mathbf{W}$ as the word embedding matrix of the neural language model since it is tied with the input word embedding matrix.

In this paper we focus on the output layer of the neural model for language generation, i.e., the softmax layer in (3.2), as in (Yang et al., 2018; Kanai et al., 2018; Ganea et al., 2019; Gao et al., 2019b). We will examine the singular value distribution of $\mathbf{W}$ and propose a different approach to increase the expressive power of the word embedding.

## 3.2 Fast Singular Value Decay

As we mentioned before, the singular values of $\mathbf{W}$ tend to drop very fast and it may interact with the cone-shaped embedding space. More specifically, Figure 1(b) and 1(d) illustrate the distributions of the normalized singular values of the Transformer-XL based language model[2] (Dai et al., 2019) and the Transformer-based machine translation model (Vaswani et al., 2017) trained on *WikiText-103* (Merity et al., 2018a) and *IWSLT 2014 De-En* (Cettolo et al., 2014) datasets, respectively. The plots show a fast singular value decay phenomenon, i.e., there is a huge drop between the first and remaining singular values. Such a phenomenon has also been observed in some previous work (Gao et al., 2019b; Wang et al., 2019a).

Figure 1(a) and 1(c) present the distributions of the projected word embeddings of the aforementioned two models. We can see from the plots that the projected word embeddings are distributed into some narrow cone shapes, which implies an anisotropic property of the learned word representations, i.e., the embedding vectors are not uniformly distributed in the space. The detailed analysis in Ethayarajh (2019) also confirms that contextualized word embeddings learnt from ELMo (Peters et al., 2018) (LSTM-based model), BERT and GPT-2 (Devlin et al., 2019; Radford et al., 2019) (Transformer-based models) indeed tend to be anisotropic.

These contextualized word embeddings have shown great success on many NLP tasks. However, static word embeddings, such as Word2Vec (Mikolov et al., 2013) and GloVe (Pennington et al., 2014) have been shown (Arora et al., 2016b; Mu & Viswanath, 2018) to be isotropic with great expressive power. Hence it may be also beneficial to increase the expressive power of the contextualized word embedding by increasing its isotropy. This motivates us to alleviate the fast singular value decay phenomenon to increase the isotropy of the learned word representations.

## 4 Proposed Method

To alleviate the fast singular value decay phenomenon, we propose to guide the singular value distribution of the contextualized word embedding throughout the training. As a result, we can achieve a trade-off between modeling contextual information, that tends to make word representations anisotropic, and the expressive power of word representations, that tends to be isotropic.

### 4.1 SVD Reparameterization

Following the previous work (Jiang et al., 2019), we propose to apply singular value decomposition (SVD) based reparameterization to the embedding matrix $\mathbf{W}$, i.e., $\mathbf{W} = \mathbf{U\Sigma V}^\top$, where $\mathbf{U} \in \mathbb{R}^{N \times d}, \mathbf{V} \in \mathbb{R}^{d \times d}$ are column orthonormal matrices, and $\mathbf{\Sigma} \in \mathbb{R}^{d \times d}$ is a diagonal matrix with $\Sigma_{kk} = \sigma_k$ being the $k$-th largest singular value of $\mathbf{W}$. Note that SVD reparameterization is standard and has been widely used in the literature such as model compression (Chen et al., 2018), training DNNs (Zhang et al., 2018), and analyzing word embeddings (Arora et al., 2016c). Given the SVD reparameterization, we can control the singular values of the embedding matrix $\mathbf{W}$ by constraining the matrix $\mathbf{E}$, and the conditional distribution in (3.2) can be rewritten as follows:

$$\mathbb{P}(\mathbf{Y}_t = \mathbf{y}_t | \mathbf{y}_{<t}) = \frac{\exp\left(\mathbf{h}_t^\top (\mathbf{U\Sigma V}^\top)_{\mathcal{I}(\mathbf{y}_t)*}\right)}{\sum_{i=1}^N \exp\left(\mathbf{h}_t^\top (\mathbf{U\Sigma V}^\top)_{i*}\right)} \text{ subject to } \mathbf{U}^\top \mathbf{U} = \mathbf{I}, \mathbf{V}^\top \mathbf{V} = \mathbf{I}, \mathbf{\Sigma} \in \mathcal{P}, \quad (4.1)$$

where $\mathcal{P}$ is a feasible set to represent the singular value distribution of the embedding matrix $\mathbf{W}$.

To ensure the orthogonal constraints in (4.1), we propose to use the orthogonal regularization for $\mathbf{U}, \mathbf{V}$ during the training process, which has been previously used in (Jiang et al., 2019). In particular, we use the following regularization, which is a linear combination of Frobenius norm and spectral norm errors: $\lambda_1 \|\mathbf{U}^\top \mathbf{U} - \mathbf{I}\|_F^2 + \lambda_2 \|\mathbf{V}^\top \mathbf{V} - \mathbf{I}\|_F^2 + \lambda_3 \|\mathbf{U}^\top \mathbf{U} - \mathbf{I}\|_2^2 + \lambda_4 \|\mathbf{V}^\top \mathbf{V} - \mathbf{I}\|_2^2$, where $\{\lambda_i\}_{i=1}^4$ are positive regularization parameters.

### 4.2 Spectrum Control

Recall the SVD of $\mathbf{W} = \mathbf{U\Sigma V}^\top$, and we consider the following two types of the singular value distribution for $\mathbf{\Sigma}$, which is inspired by the eigenvalue distribution of kernel methods (Wei et al., 2017; Pacchiano et al., 2019):

---

[2]The plots are based on the small Transformer-XL model and Transformer-Base model, the detailed model configuration can be found in experimental setups.

- **Exponential decay**: we say the singular values $\{\sigma_k\}_{k=1}^d$ of $\mathbf{W}$ satisfy the exponential decay if $\mathbf{W} \in \mathcal{P}_e(\gamma) = \{\mathbf{W} \in \mathbb{R}^{N \times d} \mid \sigma_k \leq c_1 \exp(-c_2 k^\gamma), k = 1, \ldots, d, \gamma > 0, c_1, c_2 > 0 \text{ are universal constants}\}$.

- **Polynomial decay**: we say the singular values $\{\sigma_k\}_{k=1}^d$ of $\mathbf{W}$ satisfy the polynomial decay if $\mathbf{W} \in \mathcal{P}_p(\gamma) = \{\mathbf{W} \in \mathbb{R}^{N \times d} \mid \sigma_k \leq c_1 k^{-\gamma}, k = 1, \ldots, d, \gamma > 0, c_1 > 0 \text{ is a universal constant}\}$.

For $\mathcal{P}_e(\gamma)$ and $\mathcal{P}_p(\gamma)$, the parameter $\gamma$ controls the rate of singular value decay: the larger $\gamma$ is, the faster singular value decay will be. To ensure the learned word embedding matrix $\mathbf{W}$ have the desired singular value distributions, we propose to add the following regularizations to our training objective: $\mathcal{R}_e(\mathbf{\Sigma}) = \lambda_e \sum_{k=1}^d \left( \sigma_k - c_1 \exp(-c_2 k^\gamma) \right)^2$ for exponential decay and $\mathcal{R}_p(\mathbf{\Sigma}) = \lambda_p \sum_{k=1}^d \left( \sigma_k - c_1 k^{-\gamma} \right)^2$ for polynomial decay, where $\lambda_e, \lambda_p$ are positive regularization parameters. Although the concept of "spectrum control" was previously used in Jiang et al. (2019) to improve the training of GANs, our spectrum control method is trying to solve a totally different problem, i.e., neural language generation, and its motivation is coming from a very different perspective, i.e., the representation degeneration of the word representations. In addition, our method of controlling the singular value with prior distributions is significantly different from the penalty function used in their method. Finally, our method is essential to improve the performance of the neural language generation, while the penalty function proposed in Jiang et al. (2019) can deteriorate the training of neural language models, as we illustrated in Appendix B.

### 4.3 THEORETICAL ANALYSIS

In this subsection, we show some theoretical insights on why our proposed method can improve the performance of the NLG model. In particular, we follow the similar setup as considered in (Gao et al., 2019b), i.e., focusing on the optimization of the embedding matrix $\mathbf{W} \in \mathbb{R}^{N \times d}$ and assume all the other parameters are fixed and well-optimized. In practice, we use our method to train the models from scratch. Therefore, according to the output layer in (3.2), we consider the following empirical risk minimization problem: given a training dataset $S = \{(\mathbf{h}_i, y_i)\}_{i=1}^n$ with each example drawn i.i.d. from some unknown but fixed distribution $\mathcal{D}$, and $\mathbf{h}_i \in \mathbb{R}^d$ as a hidden state, $y_i \in \{1, \ldots, N\}$ as its associated label, our goal is to minimize the training loss as follows:

$$\min_{\mathbf{W} \in \mathbb{R}^{N \times d}} L_S(\mathbf{W}) = \frac{1}{n} \sum_{i=1}^n \ell(\mathbf{h}_i^\top \mathbf{W}, y_i), \tag{4.2}$$

where $\ell(\mathbf{h}_i^\top \mathbf{W}, y_i)$ is the cross-entropy loss with respect to $\mathbf{h}_i^\top \mathbf{W}$ and $y_i$. The cross-entropy loss defined above is widely used to train NLG models, and is also used to compute the perplexity of the trained model, which is the benchmark criterion to evaluate the performance of language models.

In addition, we define the expected loss as follows $L_{\mathcal{D}}(\mathbf{W}) = \mathbb{E}[\ell(\mathbf{h}^\top \mathbf{W}, y)]$, where the expectation is taken over the distribution $\mathcal{D}$ of the training data.

Let $\widehat{\mathbf{W}} = \arg\min_{\mathbf{W} \in \mathcal{P}(\gamma)} L_S(\mathbf{W})$, where $\mathcal{P}(\gamma) = \mathcal{P}_e(\gamma)$ for exponential decay and $\mathcal{P}(\gamma) = \mathcal{P}_p(\gamma)$ for polynomial decay. We assume the right singular vector matrix satisfies $\|\mathbf{V}\|_1 \leq V$ for all $\mathbf{W} \in \mathcal{P}(\gamma)$. Now, we are ready to provide the main theory of our method (The proof can be found in Appendix A).

**Theorem 4.1.** Under previously stated conditions, suppose that $|\ell(\cdot)| \leq B$, $\ell$ is $G$-Lipschitz continuous, and $\|\mathbf{h}_i\|_\infty \leq H$ for all $i = 1, \ldots, n$. If we choose $\gamma > 1/2$, then with probability at least $1 - \delta$, we have

$$L_{\mathcal{D}}(\widehat{\mathbf{W}}) \leq \min_{\mathbf{W} \in \mathcal{P}(\gamma)} L_S(\mathbf{W}) + \frac{C_1 A N \left( \sqrt{\sum_{j=1}^{m-1} \sigma_j^2} + \sqrt{m^{1-2\gamma}/(2\gamma - 1)} \right) + C_2 B \sqrt{\log(1/\delta)}}{\sqrt{n}},$$
$$\tag{4.3}$$

where $C_1, C_2$ are absolute constants, $m \in [2, d]$, $A = GVH\sqrt{\log d}$.

**Remark 4.2.** According to Theorem 4.1, the expected loss of the learned embedding matrix $\widehat{\mathbf{W}}$ consists of two terms. The first term represents the training loss, the second term is the generalization error gap. Specifically, the smaller the $\gamma$, the larger the feasible set $\mathcal{P}(\gamma)$, thus the smaller the training loss $\min_{\mathbf{W} \in \mathcal{P}(\gamma)} L_S(\mathbf{W})$. On the other hand, the larger the $\gamma$, the faster the singular value decays, and the smaller the generalization error gap. Therefore, our generalization error bound demonstrates an appealing property of our proposed method: by directly controlling the singular value distribution

of the learned word embedding, we are able to achieve a trade-off between the training loss and generalization error.

**Remark 4.3.** For the generalization error bound in (4.3), penalizing the largest singular value could reduce this upper bound. It validates the method proposed in Gao et al. (2019b), which implicitly penalizes the largest singular value (c.f. Section 5 in Gao et al. (2019b)). Compared with their method, the error term $\widetilde{O}\big(N\sqrt{m^{1-2\gamma}/(2\gamma-1)}/\sqrt{n}\big)$ suggests that by explicitly manipulating the singular value distribution, our method has a better control of the tail sum of the singular values.

## 5 EXPERIMENTS

We demonstrate the effectiveness of our proposed spectrum control algorithm on two tasks: language modeling and machine translation. We compare our results with the state-of-the-art models. In our experiments, we try both exponential and polynomial singular value decays, and present the one with the better result. The performances of exponential decay and polynomial decay are very close (See Appendix D). In practice we found that for large scale dataset its better to use polynomial decay and for small scale dataset its better to use exponential decay. We also present the training time and memory cost of our method in Appendix C.

### 5.1 LANGUAGE MODELING

**Datasets**   We consider two benchmark datasets for language modeling: *WikiText-2* and *WikiText-103*, which consist of pre-processed Wikipedia articles and were introduced by Merity et al. (2018a). *WikiText-2* is a small dataset with around 2 million words and 30K vocabulary size, while *WikiText-103* is a significantly large dataset with around 103 million words and 260K vocabulary size.

**Model Configuration**   On the small *WikiText-2* dataset, we implement our method based on the state-of-the-art AWD-LSTM model (Merity et al., 2018a). It is a 3-layer LSTM model with 1150 dimensional hidden states and 400 dimensional embeddings. We also follow the same regularization and optimization procedures introduced in (Merity et al., 2018a). The implementation of our method is based on the open-source code[3] for AWD-LSTM. On the large *WikiText-103* dataset, we implement our method based on the state-of-the-art Transformer-XL based models (Dai et al., 2019). We follow the same settings reported in (Dai et al., 2019), and our implementation is based on the official code[4] for Transformer-XL. To evaluate the performance of our method more thoroughly, we consider two Transformer-XL models with different number of layers. The first is the standard Transformer-XL model with 16 layers used in (Dai et al., 2019). For the second, we consider a smaller Transformer-XL model with just 4 layers and other configurations unchanged.

**Parameters**   For the parameters $\{\lambda_i\}_{i=1}^4$ of the orthogonal regularizations, we tune them by grid search over $\{0.01, 0.1, 1, 10\}$. For the parameters $\lambda_e, \lambda_p$ of the spectrum control, we tune them over the grid $\{0.1, 1, 10, 100\}$. We try different singular value distributions, and the best distributions for AWD-LSTM and Transformer-XL are exponential and polynomial, respectively.

**Results of the LSTM Model on *WikiText-2***   We first present the results of language modeling on the small dataset *WikiText-2* using LSTM models. In Table 1 we compare the validation/test perplexity[5] of the baseline AWD-LSTM model (Merity et al., 2018a), the cosine similarity regularization (MLE-CosReg) model (Gao et al., 2019b), and the models trained using our method under three different settings (Merity et al., 2018a): without finetune, with finetune and with further continuous cache pointer. Compared with the baselines, our method achieves $2.3/2.9/2.3$ test perplexity reduction under all three settings; compared with the MLE-CosReg method, our method achieves $1.5/1.2/0.3$ test perplexity reduction under all three settings.

**Results of the Transformer-XL Model on *WikiText-103***   We next show the results of language modeling on the large dataset *WikiText-103* using Transformer-XL models. Table 2 compares the validation/test perplexity of Transformer-XL based models (Dai et al., 2019) and the models trained by our method on WikiText-103 dataset. The results demonstrate that our method consistently improves upon the small Transformer-XL model ($0.9$ test perplexity reduction) and the standard Transformer-XL model ($0.8$ test perplexity reduction).

---

[3]https://github.com/salesforce/awd-lstm-lm

[4]https://github.com/kimiyoung/transformer-xl

[5]Lower perplexity means better language models.

Table 1: Comparison of different methods in terms of perplexity on *WikiText-2* dataset for the task of language modeling.

| Method | Parameters | Validation | Test |
|---|---|---|---|
| Existing results | | | |
| Variational LSTM (Inan et al., 2017) | 51M | 91.5 | 87.0 |
| 2-layer skip connection LSTM (Mandt et al., 2017) | 24M | 69.1 | 65.9 |
| w/o finetune | | | |
| AWD-LSTM (Merity et al., 2018a) | 33M | 69.1 | 66.0 |
| MLE-CosReg (Gao et al., 2019b) | 33M | 68.2 | 65.2 |
| **Ours** | 33M | **66.3** | **63.7** |
| + finetune | | | |
| AWD-LSTM (Merity et al., 2018a) | 33M | 68.6 | 65.8 |
| MLE-CosReg (Gao et al., 2019b) | 33M | 67.1 | 64.1 |
| **Ours** | 33M | **65.3** | **62.9** |
| + continuous cache pointer | | | |
| AWD-LSTM (Merity et al., 2018a) | 33M | 53.8 | 52.0 |
| MLE-CosReg (Gao et al., 2019b) | 33M | 51.7 | 50.0 |
| **Ours** | 33M | **51.1** | **49.7** |

Table 2: Comparison of different methods in terms of perplexity on *WikiText-103* dataset for the task of language modeling.

| Method | Parameters | Validation | Test |
|---|---|---|---|
| Existing results | | | |
| 4 layer QRNN (Merity et al., 2018b) | 151M | 32.0 | 33.0 |
| Hebbian + Cache (Rae et al., 2018) | − | 29.7 | 29.9 |
| Small Transformer-XL (Dai et al., 2019) | 120M | 29.6 | 30.4 |
| **Ours** | 120M | **29.0** | **29.5** |
| Standard Transformer-XL (Dai et al., 2019) | 151M | 23.1 | 24.0 |
| **Ours** | 151M | **22.9** | **23.2** |

**Analysis**   We study the output embedding matrix of Transformer-XL trained on *WikiText-103* using our method. In particular, we want to evaluate the isotropy of the learned word representations. We consider the partition function $Z(\mathbf{a}) = \sum_{i=1}^{N} \exp(\langle \mathbf{w}_i, \mathbf{a} \rangle)$ introduced in (Arora et al., 2016b), where $\mathbf{w}_i$ is the $i$-th row of the embedding matrix $\mathbf{W} \in \mathbb{R}^{N \times d}$ and $\mathbf{a} \in \mathcal{S}^{d-1}$ is a unit vector. According to Lemma 2.1 in (Arora et al., 2016b), if the word representation vectors are isotropic, $Z(\mathbf{a})$ is close to some constant with high probability for all unit vectors. Thus to empirically measure the isotropy of the learned word representations, we consider two criteria based on $Z(\mathbf{a})$:

$$I_1(\mathbf{W}) = \frac{\min_{\mathbf{a} \in \mathcal{E}} Z(\mathbf{a})}{\max_{\mathbf{a} \in \mathcal{E}} Z(\mathbf{a})} \quad \text{and} \quad I_2(\mathbf{W}) = \sqrt{\frac{\sum_{\mathbf{a} \in \mathcal{E}} (Z(\mathbf{a}) - \bar{Z}(\mathbf{a}))^2}{|\mathcal{E}| \bar{Z}(\mathbf{a})^2}},$$

where $\mathcal{E}$ is the set of eigenvectors of $\mathbf{W}^\top \mathbf{W}$, as suggested by Mu & Viswanath (2018). We also propose to check sampled standard deviation measure $I_2(\mathbf{W})$ (normalized by its average, i.e., $\bar{Z}(\mathbf{a})$). We have $I_1(\mathbf{W}) \in [0, 1]$ and $I_2(\mathbf{W}) \geq 0$. Larger $I_1(\mathbf{W})$ and smaller $I_2(\mathbf{W})$ indicate more isotropic for word embeddings. We uniformly sample 40K words from the vocabulary (around 260K) of *WikiText-103* to compute these two criteria. The left half of Table 3 summarizes the values of $I_1(\mathbf{W})$ and $I_2(\mathbf{W})$, which are averaged over 10 runs, for the baseline method and our method. We can see from the results that our method significantly improves the isotropy of the learned word representations in terms of both criteria.

## 5.2   MACHINE TRANSLATION

We also apply our spectrum control method to machine translation tasks. Given a source sentence $\mathbf{s}$, the decoder of an neural machine translation (NMT) model is to predict the next word in the target sentence $\mathbf{t}$ and the previous decoded words in $\mathbf{t}$. In the following we use the state-of-the-art Transformer-based NMT model as our baseline.

Table 3: Comparison of different methods in terms of isotropy for the tasks of language modeling and machine translation. (For perfect isotropy, $I_1(\mathbf{W}) = 1$, $I_2(\mathbf{W}) = 0$.)

| Language Modeling | | | Machine Translation | | |
|---|---|---|---|---|---|
| **Method** | $I_1(\mathbf{W})$ | $I_2(\mathbf{W})$ | **Method** | $I_1(\mathbf{W})$ | $I_2(\mathbf{W})$ |
| Standard Transformer-XL | 0.24 | 0.037 | Transformer-Base | 0.31 | 0.031 |
| **Ours** | **0.63** | **0.022** | **Ours** | **0.88** | **0.005** |

Table 4: Comparison of different methods in terms of BLEU scores on the task of De→En machine translation, trained on IWSLT 2014 dataset.

| | **Method** | | | |
|---|---|---|---|---|
| *IWSLT 2014 De→En* | Adversarial (Wang et al., 2019a) | Dual-learning (Wang et al., 2019b) | Transformer-Base (Wang et al., 2019b) | **Ours** |
| **BLEU** | 35.18 | 35.44 | 34.01 | **35.50** |

**Datasets**   We compare various NMT models on the *IWSLT 2014 German→English (De-En)* and *WMT 14 English→German (En-De)* datasets. For *IWSLT 2014 De-En*, we follow the same setup as in (Gehring et al., 2017). More specifically, we have 160K sentence pairs as the training data, 7K sentence pairs as the validation data, and we combine *tst2010*, *tst2011*, *tst2012*, *dev2010* and *dev2012* datasets to form our test data. For the large scale *WMT 14 En-De*, we have 4.5 million sentence pairs as our training data, and we use *newstest2014* as our test data dataset (Cettolo et al., 2014).

**Model Configuration**   We implement our method on top of Transformer model (Vaswani et al., 2017). In particular, we consider the Transformer-Base architecture (Vaswani et al., 2017) for *IWSLT 2014 De-En*, which has a 6-layer encoder and 6-layer decoder with 512 dimensional hidden states and embeddings, except that we choose the dimension of the inner feed-forward layer as 1024 instead of 2048 and the number of attention heads is set to be 4 rather than 8. For *WMT 14 En-De*, we consider the original Transformer-Base and Transformer-Big architectures (Vaswani et al., 2017), which have a 6-layer encoder and 6-layer decoder with 512 and 1024 dimensional embeddings, respectively. Our implementation is based on the open-sourced code[6] provided by Ott et al. (2018). We follow the same procedures as in the language modeling task to choose the regularization parameters. The best results for this task on *IWSLT 2014 De-En* are from the models where the singular values are controlled by exponential distribution, and the best results for this task on *WMT 14 En-De* are from the models where the singular values are controlled by polynomial distribution.

**Results**   Comparisons of different methods in terms of BLEU scores for *IWSLT 2014 De-En* are summarized in Table 4. Compared with the baseline models, our method improves the BLEU score[7] from 34.01 to 35.50 on the German→English task, close to 1.5 gain on BLEU score; our method is also better than 35.18 reported in (Wang et al., 2019a), and 35.44 reported in (Wang et al., 2019b). Table 5 summarizes different methods in terms of BLEU scores for *WMT 14 En-De*. The results show that our method achieves 1.15 and 0.92 BLEU score improvements on this task for base and big models, respectively. In addition, our method is also better than 28.38 and 28.94 reported in Gao et al. (2019b) for base and big models.

**Analysis**   We also study the learned word embedding matrix of the Transformer trained on *IWSLT 2014 De-En* using our method in terms of isotropy. The values of $I_1(\mathbf{W})$ and $I_2(\mathbf{W})$ are reported in the right half of Table 3, we compute these two values based on all the tokens. It shows that the isotropy of the learned word representations using our method increases significantly in terms of these criteria, from very anisotropic to nearly isotropic. We also demonstrate the projected word embedding matrices and the singular value distributions of different methods in Figure 3. The plots show that the learned word representations from our method are distributed isotropically in the space, which is in contrast to the narrow cone distribution from the baseline method.

---

[6]https://github.com/pytorch/fairseq

[7]Higher BLEU score represents higher quality of the machine translation.

Table 5: Comparison of different methods in terms of BLEU scores on the task of En→De machine translation, trained on WMT 2014 dataset.

| Method | BLEU |
|---|---|
| Transformer-Base (Vaswani et al., 2017) | 27.30 |
| Transformer-Base (Gao et al., 2019b) | 28.38 |
| **Transformer-Base + Ours** | **28.45** |
| Transformer-Big (Vaswani et al., 2017) | 28.40 |
| Transformer-Big (Gao et al., 2019b) | 28.94 |
| **Transformer-Big + Ours** | **29.32** |

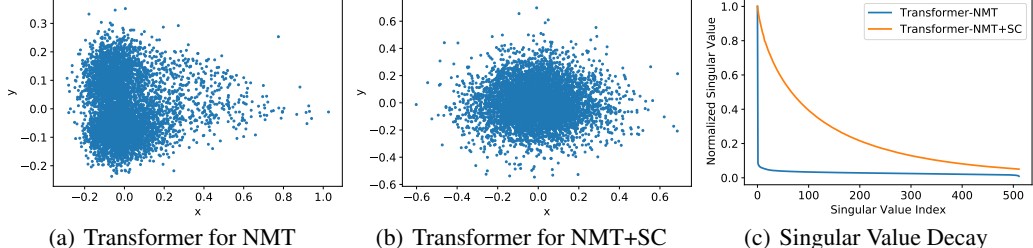

(a) Transformer for NMT    (b) Transformer for NMT+SC    (c) Singular Value Decay

Figure 3: (a) Word embedding for the vanilla Transformer, which has a narrow cone distribution; (b) Word embedding for Transformer using spectrum control, which has a uniform distribution; (c) Normalized singular value for different methods, which shows a slow decay of our method.

# 6    CONCLUSIONS AND FUTURE WORK

In this paper, we tackle the degeneration problem that occurs at the output embeddings in the softmax layer used in neural language generation models. We develop a novel spectrum control method to explicitly guide the spectra training of the output embedding matrix with some slow-decaying singular value prior distributions. Our proposed method is shown to alleviate the degeneration problem and improve isotropy of the learned contextualized word representations. Thorough experimental results demonstrate the advantage of our method over the state-of-the-art neural models for language model and machine translation. Since our work is orthogonal to Wang et al. (2019a), it would be interesting to combine the adversarial softmax training with our spectrum control method, and investigate its performance. For the future work, we would also like to investigate how the frequency-agnostic word representations in (Gong et al., 2018) relate to the single value distribution of the output embedding matrix.

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

APPENDIX

## A    PROOF OF THEOREM 4.1

**Preliminaries** We briefly review the general statistical learning framework. More specifically, let $\mathcal{X}$ and $\mathcal{Y}$ be the feature and label spaces, and suppose $\mathcal{D}$ is an unknown distribution over $\mathcal{X} \times \mathcal{Y}$. Let $\mathcal{F} \subseteq \mathcal{V}^{\mathcal{X}}$ be the hypothesis class that we use to make prediction, and $\ell : \mathcal{V} \times \mathcal{Y} \to \mathbb{R}$ be the loss function. In addition, we define the function class $\ell_{\mathcal{F}} = \{(\mathbf{x}, y) \to \ell(f(\mathbf{x}), y) : f \in \mathcal{F}\}$ as the composition of the functions in $\mathcal{F}$ and loss $\ell$. Therefore, the goal is to minimize the expected risk $L_D = \mathbb{E}_{(\mathbf{x}, y) \sim \mathcal{D}} \ell(f(\mathbf{x}), y)$ with some function $f \in \mathcal{F}$.

According to the problem setup in Section 4.3, we have $n$ i.i.d. training examples $S = \{(\mathbf{h}_i, y_i)\}_{i=1}^n$ drawn from $\mathcal{D}$, where the hidden state $\mathbf{h}_i \in \mathbb{R}^d$ is the feature vector and $y_i$ is the corresponding label. Suppose there exist a learning algorithm that maps the training dataset $S$ to a function $f \in \mathcal{F}$, and we want to measure the gap between the empirical risk $L_S$ and and the population risk $L_{\mathcal{D}}$, i.e., $|L_D - L_S|$, where $L_S = \sum_{i=1}^n \ell(f(\mathbf{h}_i), y_i)/n$. This gap is known as the generalization error.

To bound the generalization error, we use the Rademacher complexity (Bartlett & Mendelson, 2002). Let $\mathcal{F} \subseteq \mathbb{R}^{\mathcal{Z}}$ be a function class and $S = \{\mathbf{z}_1, \ldots, \mathbf{z}_n\}$ be a set of examples of size $n$, the empirical Rademacher complexity is defined as

$$\widehat{\mathfrak{R}}_S(\mathcal{F}) := \frac{1}{n} \mathbb{E}_\rho \left[ \sup_{f \in \mathcal{F}} \sum_{i=1}^n \rho_i f(\mathbf{x}_i) \right],$$

where $\rho_1, \ldots, \rho_n$ are i.i.d. Rademacher random variables with $\mathbb{P}(\rho_i = 1) = \mathbb{P}(\rho_i = -1) = 1/2$.

According to the loss function $L_S$ defined in (4.2), we are considering $N$ function classes $\{\mathcal{F}_k\}_{k=1}^N$ with $\mathcal{F}_k = \{f : \mathbf{h} \to \langle \mathbf{w}_k, \mathbf{h} \rangle, \mathbf{w}_k = \mathbf{e}^{k\top} \mathbf{W}, \mathbf{W} \in \mathcal{P}(\gamma), \mathbf{h} \in \mathbb{R}^d\}$, where $\mathbf{e}^k$ is a $d$-dimensional vector with $k$-th element to be one and others to be zero, and the loss $\ell : \mathbb{R}^N \to \mathbb{R}$. We have the following Lemma to bound the generalization error based on the empirical Rademacher complexity, which was proved in Corollary A.11 in Allen-Zhu et al. (2019).

**Lemma A.1.** (Allen-Zhu et al., 2019) If $\mathcal{F}_1 \ldots, \mathcal{F}_N$ are N classes of functions $\mathbb{R}^d \to \mathbb{R}$, the loss function $\ell : \mathbb{R}^N \to \mathbb{R}$ is $G$-Lipschitz continuous and $|\ell(\cdot)| \leq B$ for any $\mathbf{z} \sim \mathcal{D}$, then with probability at least $1 - \delta$, we have

$$\sup_{f_1 \in \mathcal{F}_1, \ldots, f_n \in \mathcal{F}_N} \left| \mathbb{E}\big[\ell\big(f_1(\mathbf{z}), \ldots, f_N(\mathbf{z})\big)\big] - \frac{1}{n} \sum_{i=1}^n \ell(f(\mathbf{z}_i)) \right| \leq C_1 G \sum_{k=1}^N \widehat{\mathfrak{R}}_S(\mathcal{F}_k) + C_2 \frac{B\sqrt{\log(1/\delta)}}{\sqrt{n}},$$

where the expectation is taken over the distribution $\mathcal{D}$.

Equipped with this lemma, we now present the proof of our main result.

*Proof of Theorem 4.1.* According to the problem setup in Section 4, we have $N$ classes of functions $\{\mathcal{F}_k\}_{k=1}^N$ and let the singular value decomposition of $\mathbf{W}$ as

$\mathbf{W} = \mathbf{U}\mathbf{\Sigma}\mathbf{V}^\top$ and $\mathbf{u}_k$ is the $k$-th row of $\mathbf{U}$. Therefore, we have

$$\widehat{\mathfrak{R}}_S(\mathcal{F}_k) = \mathbb{E}_\rho\left[\sup_{f\in\mathcal{F}_k} \frac{1}{n}\sum_{i=1}^n \rho_i\langle\mathbf{w}_k, \mathbf{h}_i\rangle\right] = \mathbb{E}_\rho\left[\sup_{f\in\mathcal{F}_k} \frac{1}{n}\langle\mathbf{V}\mathbf{\Sigma}\mathbf{u}_k^\top, \mathbf{h}_\rho\rangle\right] \tag{A.1}$$

where $\mathbf{h}_\rho = \sum_{i=1}^n \rho_i\mathbf{h}_i$ and the last inequality is due to the definition of $\mathbf{w}_k = \mathbf{V}\mathbf{\Sigma}\mathbf{u}_k^\top$. Therefore, we have

$$\widehat{\mathfrak{R}}_S(\mathcal{F}_k) \leq \mathbb{E}_\rho\left[\sup_{f\in\mathcal{F}_k} \frac{1}{n}\|\mathbf{V}\mathbf{\Sigma}\mathbf{u}_k^\top\|_1\|\mathbf{h}_\rho\|_\infty\right]$$

$$\leq \frac{V}{n}\mathbb{E}_\rho\left[\sup_{f\in\mathcal{F}_k} \|\mathbf{\Sigma}\mathbf{u}_k^\top\|_1\|\mathbf{h}_\rho\|_\infty\right]$$

$$\leq \frac{V\sqrt{\sum_{j=1}^d \sigma_j^2}}{n}\mathbb{E}_\rho\left[\sup_{f\in\mathcal{F}_k} \|\mathbf{h}_\rho\|_\infty\right], \tag{A.2}$$

where the second inequality is due to $\|\mathbf{V}\|_1 \leq V$ and the last one comes from $\|\mathbf{\Sigma}\mathbf{u}_k^\top\|_1 = \sum_{j=1}^d \sigma_j|u_{kj}| \leq \sqrt{\sum_{j=1}^d \sigma_j^2}$. In addition, according to Theorem 12 in Liang (2014), we have

$$\mathbb{E}_\rho[\|\mathbf{h}_\rho\|_\infty] \leq H\sqrt{2n\log d},$$

Therefore, we can get

$$\widehat{\mathfrak{R}}_S(\mathcal{F}_k) \leq VH\frac{\sqrt{\sum_{i=1}^m \sigma_i^2} + \sqrt{\sum_{i>m} \sigma_i^2}}{\sqrt{n}}\sqrt{2\log d}.$$

As a result, we can get

$$\sum_{k=1}^N \widehat{\mathfrak{R}}_S(\mathcal{F}_k) \leq NVH\frac{\sqrt{\sum_{i=1}^m \sigma_i^2} + \sqrt{\sum_{i>m} \sigma_i^2}}{\sqrt{n}}\sqrt{2\log d}. \tag{A.3}$$

Since for $\mathcal{P}(\gamma) = \mathcal{P}_e(\gamma)$, we have $\sigma_j \leq c_1\exp(-c_2 j^\gamma) \leq c_1/(c_2 j^\gamma)$, and for $\mathcal{P}(\gamma) = \mathcal{P}_p(\gamma)$, we have $\sigma_j \leq c_3 j^{-\gamma}$. Thus for $\gamma > 1/2$, we have

$$\sum_{j>m} \sigma_j^2 \leq \sum_{j>m} \frac{c_4}{j^{2\gamma}} \leq c_4\int_{m+1}^\infty x^{-2\gamma}dx \leq \frac{c_4}{(2\gamma-1)}(m+1)^{1-2\gamma},$$

where $c_4 = (c_1/c_2)^2$ if $\mathcal{P}(\gamma) = \mathcal{P}_e(\gamma)$ and $c_4 = c_3^2$ if $\mathcal{P}(\gamma) = \mathcal{P}_p(\gamma)$.

Therefore, plugging this upper bound into (A.3), we have

$$\sum_{k=1}^N \widehat{\mathfrak{R}}_S(\mathcal{F}_k) \leq \frac{NVH\sqrt{2\log d}\sqrt{\sum_{j=1}^m \sigma_j^2}}{\sqrt{n}} + c_5\frac{NVH\sqrt{2\log d}\sqrt{m^{1-2\gamma}/(2\gamma-1)}}{\sqrt{n}}.$$

Where $c_5 = c1/c5$ if $\mathcal{P}(\gamma) = \mathcal{P}_e(\gamma)$ and $c_5 = c_3$ if $\mathcal{P}(\gamma) = \mathcal{P}_p(\gamma)$.

According to Lemma A.1, since $\ell$ is $G$-Lipschitz continuous and $|\ell(\cdot)| \leq B$, we have

$$\sup_{\mathbf{W}\in\mathcal{P}(\gamma)} |L_\mathcal{D}(\mathbf{W}) - L_S(\mathbf{W})| \leq c_6 GNVH\sqrt{\log d}\frac{\sqrt{\sum_{j=1}^m \sigma_j^2} + \sqrt{\frac{m^{1-2\gamma}}{(2\gamma-1)}}}{\sqrt{n}}$$

$$+ c_7 B\sqrt{\frac{\log(1/\delta)}{n}}.$$

$\square$

Table 6: Comparison of different penalty functions in terms of perplexity on *WikiText-2* dataset.

| Method | Validation | Test |
|---|---|---|
| AWD-LSTM (Merity et al., 2018a) | 69.1 | 66.0 |
| Spectrum Normalization | 76.3 | 72.5 |
| SN+D-Optimal Regularizer | 98.3 | 94.8 |
| **Ours** | **66.3** | **63.7** |

## B    EXPERIMENTAL RESULTS FOR OTHER PENALTY FUNCTIONS

In this section, we also implement our method with the penalty function proposed in Jiang et al. (2019) for training GANs. We consider *WikiText-2* dataset, and all the experimental settings are same as before. Table 6 demonstrates the performance of our method using the Spectrum Normalization (SN) and SN+D-Optimal Regularizer, which can achieve the best performance of training GANs in their paper. The results show that their proposed penalty function can deteriorate the training of neural language models. This is because their method is motivated from training GANs, and will encourage all the singular values close to the largest one. If we use such penalty function to train neural language models, the learned word representations will lose the power of modeling contextual information. Therefore, our proposed spectrum control method is essential to improve the training of neural language generation.

## C    TRAINING TIME AND MEMORY COST

In this section, we compare the training time and memory cost of our method with the baseline method, i.e., AWD-LSTM (Merity et al., 2018a) and standard Trandformer-XL (Dai et al., 2019). More specifically, we test our method and the baseline method on the same machine on *WikiText-2 WikiText-103*, and *WMT 14* datasets. For *WikiText-2* dataset, we use one NVIDIA Tesla V100 GPU and set the batch size to be 80. For *WikiText-103* dataset, we use four NVIDIA Tesla V100 GPU and set the batch size to be 40. For *WMT 14*, we use four NVIDIA Tesla V100 GPU and set the max token as 3500. The training time and memory cost for a single GPU are summarized in Table 7. The results illustrate that our method is only $1.17\times$, $1.18\times$, and $1.24\times$ slower than the baseline methods on *WikiText-2*, *WikiText-103* and *WMT 14* datasets, respectively. In addition, our method will cost $1.06\times$, $1.32\times$ and $1.11\times$ memory compared with the baseline methods on *WikiText-2*, *WikiText-103*, and *WMT 14* datasets, respectively.

Table 7: Comparisons of our method and baseline method in terms of the average training time per epoch and the memory cost.

| Method | *WikiText-2* | *WikiText-103* | *WMT 14* |
|---|---|---|---|
| Training time | | | |
| Baseline | 61 sec | 2681 sec | 5991 sec |
| Ours | 70 sec | 3152 sec | 7421 sec |
| Memory cost | | | |
| Baseline | 8.9 GB | 11.4 GB | 14.2 GB |
| Ours | 9.5 GB | 15.0 GB | 15.7 GB |

Table 8: Comparisons of different priors on different datasets.

| Dataset | Exponential Decay | Polynomial Decay |
|---|---|---|
| Language Modeling | Test PPL | Test PPL |
| *WikiText-2* | **63.7** | 64.2 |
| *WikiText-103* | 23.4 | **23.2** |
| Machine Translation | BLEU | BLEU |
| *IWSLT 2014 De→En* | **35.50** | 35.40 |
| *WMT 14 En→De* | 28.37 | **28.45** |

## D    COMPARISON OF TWO PRIORS

In this section we present the performance of our method on different tasks using different prior distributions. Table 8 suggests that the overall performance of these two prior distributions on different tasks are similar. In addition, the results show that it is better to use polynomial decay for large scale dataset and exponential decay for small scale dataset.

