# OpenReview forum: "Improving Neural Language Generation with Spectrum Control"
_ICLR.cc/2020/Conference — Accept (Poster)_

### Official Review · AnonReviewer2 · 2019-10-21
**Official Blind Review #2**

**Rating:** 6

**Review:**

Summary: This paper deals with the representation degeneration problem in neural language generation, as some prior works have found that the singular value distribution of the (input-output-tied) word embedding matrix decays quickly. The authors proposed an approach that directly penalizes deviations of the SV distribution from the two prior distributions, as well as a few other auxiliary losses on the orthogonality of U and V (which are now learnable). The experiments were conducted on small and large scale language modeling datasets as well as the relatively small IWSLT 2014 De-En MT dataset.

Pros:
+ The paper is well-written with great clarity. The dimensionality of the involved matrices (and their decompositions) are clearly provided, and the approach is clearly described. The authors also did a great job providing the details of their experimental setup.
+ The experiments seem to show consistent improvements over the baseline methods (at least the ones listed by the authors) on a relatively extensive set of tasks (e.g., of both small and large scales, of two different NLP tasks). Via WT2 and WT103, the authors also showed that their method worked on both LSTM and Transformers (which it should, as the SVD on word embedding should be independent of the underlying architecture).
+ I think studying the expressivity of the output embedding matrix layer is a very interesting (and important) topic for NLP. (e.g., While models like BERT are widely used, the actual most frequently re-used module of BERT is its pre-trained word embeddings.)

---------------------------------

I have a few questions/comments on the work as well:

1) One of the things that is not clearly described in the paper is how the proposed spectrum control method was injected into training in practice. In Section 4.3 (when you do the theoretical analysis), you assumed "all the other parameters are fixed and well-optimized". Is that also what you did in training on WT2/WT103 (e.g., first pre-train a model such as Transformer-XL, and then fine-tune its embedding layer using the proposed decomposed method)?

2) How does the runtime and memory cost of your approach compare to the baselines? (e.g., you now need to compute $U^\top U$, which can also be prohibitively large when the vocabulary size is large; for instance, on the 1-Billion word dataset).

3) I remember the base Transformer-XL model did not use the adaptive embedding/softmax (although they set the `--adaptive` flag, they did use `--div_val 1`). How does your method work in the adaptive setting, where the embedding size $d$ of different words could be very different (e.g., depending on the word frequency)?

4) Regarding the theoretical analysis in Section 4.3. Why is the cross-entropy loss function (essentially NLL loss + softmax) bounded (as is required by Theorem 4.1)? Given a fixed ground-truth $y_i$, if the corresponding predicted likelihood is small (e.g., $\rightarrow 0$), won't the CE loss be very large? In that case, the generalization bound in Eq. (4.3) would be vacuous, as B would be very large too. Moreover, I'm also not fully convinced by what the theoretical analysis is trying to convey--- that your method "achieve a trade-off between the training loss and generalization error"--- but isn't that what (any) regularizations are designed for? The proved bound is no different in nature from any generalization bound with a regularizer (e.g., weight decay), and it does not necessarily reflect the usefulness of the proposed approach.

5) In experiments, you only showed the better performance of the exponential and the polynomial singular value decays (cf. beginning of Sec. 5). Could you show both? Which one is better (and on what task), and by how much? If one is to use your method, which decay scheme do you recommend?

6) Why is MLE-CosReg (Gao et al. 2019b) not compared to in the WikiText-103 and the machine translation task? As the MLE-CosReg approach only involves regularizing the cosine similarity via $\text{Sum}(\hat{W}\hat{W}^\top)$, it should computationally be even slightly cheaper than the proposed method (you need to compute $\mathbf{U}^\top \mathbf{U}$, which has the same complexity). They also tested on the larger-scale WMT En-De and De-En dataset (which contains 4.5M sentence pairs). Is there any reason that you chose IWSLT 2014 instead?

---------------------------------

Some issues that didn't impact the score:

7) It'd be useful to add labels to the x- and y-axis of the plots in Figure 1.

8) When implementing the method described in Section 4.2, did you explicitly sort the singular values in each iteration, or just set them to learnable parameters (along with learnable $\mathbf{U}, \mathbf{V}$) without sorting? If you do sort, do you also "sort" the columns of $\mathbf{U}$ and $\mathbf{V}$ (as I would expect a one-to-one mapping from $\sigma_i$ to $\mathbf{U}_i$, for instance)? If you don't sort, how do you make sure that $\sigma_i \geq \sigma_{i+1}$?

9) Why did you regularize $\mathbf{U}$ and $\mathbf{V}$ by both its Frobenius norm and its spectral norm? Does using only one of them compromise the performance?

---------------------------------

Overall, I find this work well-written and well-motivated. The experiments seem to show consistent improvement when using the approach. I vote for weak accept, but I also look forward to the author's response to the questions I raised above.

**Experience Assessment:**

I have published one or two papers in this area.

**Review Assessment: Checking Correctness Of Derivations And Theory:**

I assessed the sensibility of the derivations and theory.

**Review Assessment: Checking Correctness Of Experiments:**

I carefully checked the experiments.

**Review Assessment: Thoroughness In Paper Reading:**

I read the paper thoroughly.

---

> ### Author Response · Authors · 2019-11-10
> **Response to Reviewer #2**
>
> Thanks for your constructive comments, we answer your questions/comments as follows:
>
> 1) In practice we use our method to train the models from scratch, we have clarified this in Section 4.3 in the revision.
>
> 2) Our method is efficient and the memory cost is reasonable, and we have added a training time and memory cost comparison in Table 7 in Appendix D. On the WikiText-2 and WikiText-103 datasets, our method is only $1.17\times$ and $1.18\times$ slower than the baseline methods, respectively. Note that our method is more efficient than the method proposed by Gao et al. 2019b. Because the computational complexity for our method to compute $\mathbf{U}^\top\mathbf{U}$ is $O(Nd^2)$ while the computational complexity for Gao et al. 2019b to compute $\hat W \hat W^\top$ is $O(N^2d)$, where $N$ is the vocabulary size and $d$ is the embedding dimension ($N$ is often much larger than $d$). Since our method will only have slightly larger number of parameters ($O(d^2)$) than the baseline method due to the SVD reparameterization and the regularizations, the extra memory cost is reasonable. On the WikiText-2 and WikiText-103 datasets, our method will cost $1.06\times$ and $1.32\times$ memory than the baseline methods, respectively.
>
> 3) We did not use the adaptive embedding/softmax in our method as in the base Transformer-XL model setting. To apply our method to the adaptive setting, we think it is reasonable to choose different embedding sizes $d$ following the original adaptive method.
>
> 4) In Theorem 4.1 we directly assume that cross entropy loss is bounded when the embedding matrix $\mathbf{W}$ belongs to a certain constraint $\mathcal{P}(\gamma)$. It is true that $B$ can be large since it is a universal upper bound that covers the worst case scenario. In practice, $B$ can be small if the cross entropy loss is well optimized. Having a dependence on $B$ is the limitation of the uniform convergence based analysis for generalization bound, and can be addressed by a more careful analysis such as algorithm dependent based analysis. Nevertheless, this is beyond the scope of this paper. On the other hand, our theoretical result explicitly shows the effect of singular values on the expected loss, which is different from the generalization bound with a dependence on the norm of weight matrices. Our bound in Theorem 4.1 sheds light on the advantages of our method by directly controlling the singular value distribution.
>
> 5) The performances of exponential decay and polynomial decay are very close, and we have added their comparisons in Appendix E. For example, on WikiText-103, the test PPL for exponential decay is 23.4 compared with 23.2 reported in the paper for polynomial decay. In practice we found that for large scale dataset it’s better to use polynomial decay and for small scale dataset it’s better to use exponential decay. We have added the above comments at the beginning of the experiment section.
>
> 6) Since the authors of Gao et al. 2019 do not provide the source code of their method, we did not compare with their method on these two datasets. As we mentioned before, the computational complexity of our method to compute $\mathbf{U}^\top\mathbf{U}$ is less than their method to compute $\hat W \hat W^\top$. We are running experiments on the large-scale WMT 14 En-De dataset as you suggested. Currently our method can achieve 28.45/29.32 BLEU scores for Transformer base/ big models, respectively, which are better than 28.38/28.94 BLEU scores reported in Gao et al. 2019b (see Table 5 in Appendix B). We will report the final results of our method once we finish the training.
>
> 7) We have added the labels for x- and y-axis in Figure 1 in the revision.
>
> 8) In our implementation, we just set singular values to learnable parameters. As a result, the learned singular values are not automatically sorted. When we implement the spectrum control, we will sort the singular values to achieve the regularization for the singular value distribution. In this way, we do not need “sort” the columns of $\mathbf{U},\mathbf{V}$ or make sure $\sigma_i\geq \sigma_{i+1}$.
>
> 9) In practice, we can use any norm based error, such as $\ell_\infty$-norm, $\ell_2$-norm, Frobenius norm, and their combinations. In practice we found that the combination of $\ell_2$ and Frobenius norms can give us slightly better results than only use one, and that’s why we choose this combination.

---

### Official Review · AnonReviewer3 · 2019-10-23
**Official Blind Review #3**

**Rating:** 3

**Review:**

Authors propose to apply Spectrum control regularization to the embedding of weight matrices in NLP problems such as language modeling and neural machine translation. Spectrum Control Regularization was originally proposed and applied to GANs (Jiang et al 2019)

The author motivate the approach by showing that the singular values of embedding weight matrices, although I am not convinced that it is such a big issue. In terms of experimental results authors show a very slight improvement over strong baseline models, that further shows an evidence that regularization singular values of embedding matrices is not very important.

Overall the paper is written well, however the contribution is very marginal.


**Experience Assessment:**

I have published in this field for several years.

**Review Assessment: Checking Correctness Of Derivations And Theory:**

I assessed the sensibility of the derivations and theory.

**Review Assessment: Checking Correctness Of Experiments:**

I assessed the sensibility of the experiments.

**Review Assessment: Thoroughness In Paper Reading:**

I read the paper at least twice and used my best judgement in assessing the paper.

---

> ### Author Response · Authors · 2019-11-10
> **Response to Reviewer #3**
>
> Thank you for your feedback on our work.
>
> Q1: “Spectrum Control Regularization was originally proposed and applied to GANs”
>
> R1: Although spectrum control has been previously used in training GANs, using it in the setting of training neural language models is arguably novel. Furthermore, our method of controlling the singular value using prior distributions is different from Jiang et al. 2019, and is essential to improve the performance of neural language generation. Directly applying their penalty functions for training GANs can deteriorate the training of neural language models. For example, the test PPL of LSTM on the WikiText-2 dataset using their best penalty function D-optimal Reg is 94.8, which is even worse the the baseline method with 66.0 test PPL (the lower PPL, the better). This is because their method will encourage all the singular values close to the largest one, and learned word representations will lose the power of modeling contextual information. We have added this additional comparison result in Table 6 in Appendix C for the training of LSTM. We will add results for training Transformer-based models if time allows.
>
>
> Q2: “The author motivate the approach by showing that the singular values of embedding weight matrices, although I am not convinced that it is such a big issue.”
>
> R2: The singular values are indeed crucial to the expressive power of the embedding matrix of the neural language model, and therefore are very important for many down streaming NLP tasks. Previous studies (Gao et al. 2019b, Ethayarajh 2019) have pointed out that the contextualized embedding matrix learned end-to-end are anisotropic, while the isotropic property of the static word embedding has been shown to be very beneficial to their expressive power (Arora et al. 2016,  Mu & Viswanath, 2018). In addition, controlling the singular value decay of the static word embedding can encourage its isotropic property (Mu & Viswanath, 2018) and thus increases its expressive power and benefits the downstream task performance. All these studies suggest that the singular values of embedding weight matrices are critical for improving its expressive power. In addition, as confirmed by our experiments, our proposed Spectrum Control Regularization can indeed promote the isotropy of the contextualized embedding matrix (shown in Figure 3 and Table 3), and further improve the performance on language modeling and machine translation.
>
>
> Q3: “In terms of experimental results authors show a very slight improvement over strong baseline models”
>
> R3: The improvements of our method are justified since our method improves 0.8 test PPL over the state-of-the-art standard Transformer-XL model on language modeling for the first time, and 1.5 BLEU score over Transformer model on machine translation. Note that they are very strong baselines and achieving such improvements is non-trivial and should be considered to be significant. We show that by only regularizing the output embedding matrix using our method can already achieve such improvements. In addition, our method can be combined with other methods that focus on the encoding layers to further improve neural language generation, which will be our future work.
>
>
> Q4: “the contribution is very marginal”
>
> R4: We would like to emphasize the contributions of our paper as follows: (1) a novel spectrum control method, i.e., using two prior distributions for controlling the singular value distribution, to improve the training of neural language models, which is different from previous spectrum control method for training GANs, and is motivated from a very different perspective; (2) extensive experiments to validate the generality and effectiveness of our proposed method; (3) a theoretical analysis to justify our proposed method; and (4) a thorough analysis of promoting isotropy of contextualized word embedding with our method, which has only been studied for the static word embedding before. We believe the contributions of our work are definitely not marginal.

---

### Official Review · AnonReviewer1 · 2019-10-23
**Official Blind Review #1**

**Rating:** 6

**Review:**

The paper proposes a regularizer for the output representation of transformer NNs, based on the singular value distribution to encourage learning of richer representations and avoid fast decay of singular values previously reported for NNs with softmax outputs.
In particular, the embedding matrix is parametrized as the product of a matrix U, a diagonal matrix Sigma and a matrix V. U and V are encouraged towards orhogonality using additional penalties similar to Lagrangian augmentation. Finally, a desired singular value distribution (exponential or polynomial decay) is encouraged by adding an appropriate regularization penalty on the entries of Sigma.

The authors present a generalization error bound that relates expected loss, training loss and singular value distribution to motiveate the choice of the regularizer.
Experiments are provided for a machine translation and languate modeling, showing mild improvements of the proposed regularziaer over the state-of-the-art baselines.

The paper is well written, notation is clearly introduced and used in consistent manner, mathematical derivations are clear and easy to follow.



**Experience Assessment:**

I have read many papers in this area.

**Review Assessment: Checking Correctness Of Derivations And Theory:**

I assessed the sensibility of the derivations and theory.

**Review Assessment: Checking Correctness Of Experiments:**

I assessed the sensibility of the experiments.

**Review Assessment: Thoroughness In Paper Reading:**

I read the paper at least twice and used my best judgement in assessing the paper.

---

> ### Author Response · Authors · 2019-11-10
> **Response to Reviewer #1**
>
> Thanks for your positive comments. Our method mainly focuses on the output layer of neural language model, and is built on top of the state-of-the-art baselines. Therefore, we believe the improvements ( 0.8 test PPL over the state-of-the-art standard Transformer-XL model,  and 1.5 BLEU over the Transformer model) achieved solely by regularizing the output layer using our method are significant. To the best of our knowledge, our work is the first one to show improvement over the state-of-the-art standard Transformer-XL LM model (Dai et al. ACL 2019).

---

### Public Comment · ~Richardo_Del_Potror1 · 2019-10-31
**Difference between Jiang et al. ICLR 2019 and yours?**

On Page 3, you mentioned that your proposed method is "inspired by the spectrum control that encourages slow singular value decay in Generative Adversarial Network (GAN) training (Jiang, 2019)".

However, when I compared your paper with Jiang, 2019

https://openreview.net/pdf?id=rJNH6sAqY7),

I found that your proposed method is just a very incremental tweak of Jiang. 2019. Your sections 4.1 and 4.2 are just Sections 2.1 and 2.2 in Jiang, 2019. You only changed a bit on the penalty of the singular values. I guess your change is not essential. The penalty function proposed in  Jiang, 2019 should also be able to improve the training.

Throughout your Sections 4.1 and 4.2, you never mentioned that all these have appeared in Jiang et al. 2019. In a nut shell, you basically copied the spectrum control method in Jiang, 2019, and claimed it as something new.

---

> ### Author Response · Authors · 2019-10-31
> **Re: Difference between Jiang et al. ICLR 2019 and yours?**
>
> Although the concept of “spectrum control” was used both in our paper and Jiang et al.’s paper, the problems studied in both papers are totally different: our paper studies neural language generation, while Jiang et al. study training GAN. Therefore, our motivation of using spectral control is coming from a very different perspective, as we stated in the introduction section and illustrated in Figures 1 and 2.
>
> To achieve the goal of spectrum control efficiently, we also propose to use the SVD reparameterization as in Jiang et al., 2019. Note that SVD reparameterization in Section 4.1 is standard and has been widely used in the literature such as model compression, training DNNs, matrix completion, and analyzing word embeddings. We don’t tweak or copy anything from Jiang et al. 2019.
>
> Furthermore, the method of controlling the singular value distribution we proposed in Section 4.2 is different from that in Jiang et al. 2019, and is essential to improve the performance of the neural language generation. In fact, the penalty function proposed in Jiang et al.’s paper can deteriorate the training of neural language models. To see this, our proposed prior distributions as shown in Figure 2 in our paper are fundamentally different from the singular value distributions learned using their penalty functions (See Figure 1 and Table 7 in Jiang et al.’s paper). Figure 1 in their paper suggests that their penalty function, i.e., D-optimal Reg, will encourage all the singular values close to 1, which is well aligned with their motivation for training GAN. However, if we use such penalty function to train neural language models, the learned word representations will lose the power of modeling contextual information, and can result in much worse results than the baseline methods.
>
> We will emphasize the key differences mentioned above in Section 4.1 and Section 4.2 in the revision during the author response phase.

---

### Decision · Program_Chairs · 2019-12-19

**Decision:**

Accept (Poster)

**Comment:**

Main content:

Blind review #2 summarizes it well:

Summary: This paper deals with the representation degeneration problem in neural language generation, as some prior works have found that the singular value distribution of the (input-output-tied) word embedding matrix decays quickly. The authors proposed an approach that directly penalizes deviations of the SV distribution from the two prior distributions, as well as a few other auxiliary losses on the orthogonality of U and V (which are now learnable). The experiments were conducted on small and large scale language modeling datasets as well as the relatively small IWSLT 2014 De-En MT dataset.

Pros:
+ The paper is well-written with great clarity. The dimensionality of the involved matrices (and their decompositions) are clearly provided, and the approach is clearly described. The authors also did a great job providing the details of their experimental setup.
+ The experiments seem to show consistent improvements over the baseline methods (at least the ones listed by the authors) on a relatively extensive set of tasks (e.g., of both small and large scales, of two different NLP tasks). Via WT2 and WT103, the authors also showed that their method worked on both LSTM and Transformers (which it should, as the SVD on word embedding should be independent of the underlying architecture).
+ I think studying the expressivity of the output embedding matrix layer is a very interesting (and important) topic for NLP. (e.g., While models like BERT are widely used, the actual most frequently re-used module of BERT is its pre-trained word embeddings.)

--

Discussion:

The reviewers agree that it is a very well written paper, and this is important as a conference paper to illuminate readers.

The one main objection is that spectrum control regularization was previously proposed and applied to GANs (Jiang et al ICLR 2019). However the authors convincingly point out that the technique is widely used, not only for GANs, and that application to neural language generation has quite different characteristics requiring a different, new approach: "our proposed prior distributions as shown in Figure 2 in our paper are fundamentally different from the singular value distributions learned using their penalty functions (See Figure 1 and Table 7 in Jiang et al.’s paper). Figure 1 in their paper suggests that their penalty function, i.e., D-optimal Reg, will encourage all the singular values close to 1, which is well aligned with their motivation for training GAN. However, if we use such penalty function to train neural language models, the learned word representations will lose the power of modeling contextual information, and can result in much worse results than the baseline methods."

--

Recommendation and justification:

I concur with the majority of reviewers that this paper is a weak accept. Though not revolutionary, it is well written, has usefully broad application, and is supported well empirically.